# Activating Transcription Factor 3 Diminishes Ischemic Cerebral Infarct and Behavioral Deficit by Downregulating Carboxyl-Terminal Modulator Protein

**DOI:** 10.3390/ijms24032306

**Published:** 2023-01-24

**Authors:** Mei-Han Kao, Chien-Yu Huang, Wai-Mui Cheung, Yu-Ting Yan, Jin-Jer Chen, Yuan-Soon Ho, Chung Y. Hsu, Teng-Nan Lin

**Affiliations:** 1Taiwan International Graduate Program in Molecular Medicine, National Yang-Ming University and Academia Sinica, Taipei 11529, Taiwan; 2Institute of Biomedical Sciences, Academia Sinica, Taipei 11529, Taiwan; 3Department of Surgery, Shuang Ho Hospital, Taipei Medical University, New Taipei City 23561, Taiwan; 4School of Medical Laboratory Science and Biotechnology, College of Medical Science and Technology, Taipei Medical University, Taipei 11031, Taiwan; 5Graduate Institute of Biomedical Sciences, China Medical University, Taichung 404327, Taiwan

**Keywords:** stroke, apoptosis, Akt/PKB, ATF3, CTMP, gene regulation

## Abstract

Activating transcription factor 3 (ATF3) is a stress-induced transcription factor and a familiar neuronal marker for nerve injury. This factor has been shown to protect neurons from hypoxic insult in vitro by suppressing carboxyl-terminal modulator protein (CTMP) transcription, and indirectly activating the anti-apoptotic Akt/PKB cascade. Despite prior studies in vitro, whether this neuroprotective pathway also exists in the brain in vivo after ischemic insult remains to be determined. In the present study, we showed a rapid and marked induction of ATF3 mRNA throughout ischemia-reperfusion in a middle cerebral artery (MCA) occlusion model. Although the level of CTMP mRNA was quickly induced upon ischemia, its level showed only a mild increase after reperfusion. With the gain-of-function approach, both pre- and post-ischemic administration of Ad-ATF3 ameliorated brain infarct and neurological deficits. Whereas, with the loss-of-function approach, ATF3 knockout (KO) mice showed bigger infarct and worse functional outcome after ischemia. In addition, these congenital defects were rescued upon reintroducing ATF3 to the brain of KO mice. ATF3 overexpression led to a lower level of CTMP and a higher level of p-Akt(473) in the ischemic brain. On the contrary, ATF3 KO resulted in upregulation of CTMP and downregulation of p-Akt(473) instead. Furthermore, post-ischemic CTMP siRNA knockdown led to smaller infarct and better behaviors. CTMP siRNA knockdown increased the level of p-Akt(473), but did not alter the ATF3 level in the ischemic brain, upholding the ATF3→CTMP signal cascade. In summary, our proof-of-principle experiments support the existence of neuroprotective ATF3→CTMP signal cascade regulating the ischemic brain. Furthermore, these results suggest the therapeutic potential for both ATF3 overexpression and CTMP knockdown for stroke treatment.

## 1. Introduction

Activating transcription factor (ATF) belongs to the basic region–leucine zipper (bZIP) motif superfamily of transcription factors. The basic region is responsible for specific DNA binding, and the leucine zipper region is involved in both DNA binding and dimerization of ATF with other bZIP-containing proteins, including ATF, CREB/CREM, AP1 (Fos/Jun), C/EBP, and Maf families [1]. In general, homodimer of ATF leads to transcriptional repression, while heterodimer activates transcription [1,2]. The ATF family contains seven members, ATF1–7, and they share high similarity within the bZIP motif, but not outside the bZIP domain [1]. Among these isoforms, ATF3, also known as LRF-1, LRG-21, CRG-5, and TI-241, has been most commonly studied. ATF3 cDNA was originally isolated in 1989 [2]. Human ATF3 gene is located on chromosome 1, spans ~56 kb, and comprises 6 exons and 2 promoters. ATF3 encodes a major isoform of 181 amino acids-22 kD protein. Unlike other ATF family genes which are regulated at the translational level, ATF3 is regulated at the transcriptional level. Its steady-state mRNA level is virtually undetectable, but greatly increases upon stress stimulations, such as H_2_O_2_, TNFα, LPS, and hypoxia [1,3,4]. ATF3 has been demonstrated to play a role in a variety of biological processes at both cellular and organismal levels, and functions in a context-dependent manner. Interestingly, the consequences of ATF3 deficiency to the host can be either beneficial or detrimental [5,6,7,8].

ATF3 is a known neuronal marker of nerve injury [9,10]. Following permanent cerebral ischemia, a biphasic induction of ATF3 has been reported: the early, transient and higher expression was noted in the ischemic core area, whereas the late, prolonged and lower induction was noted in the penumbral area [11]. Furthermore, in a rat cerebral thrombosis model, ATF3 is increased corresponding to the severity of neurological deficit and infarct size [12]. Immuno-reactivity of the induced ATF3 has been found to co-localize with the damage-induced neuronal endopeptidase (DINE), which is known to promote anti-oxidant activity and thus acts as a neuro-protective molecule under ischemic insult [13]. Following transient cerebral ischemia, induced ATF3 has been found to co-localize with apoptosis executor caspase 3 but not Bcl-xL or Hsp27 [14]. A miR-221-3p protects ischemic neurons by targeting ATF3 [15]. On the contrary, hypoxia preconditioning was shown to cause a marked induction of ATF3 and subsequently reduced infarct in the ischemic brain [16]. There is evidence that Hsp27 and tissue plasminogen activator (tPA) could attenuate ischemic damage via activation of ATF3 [17,18,19]. Nuclear calcium-CREB-ATF3 signaling and pre-ischemic ATF3 overexpression by recombinant adeno-associated virus (rAAV) has been shown to reduce NMDA-induced brain damage [20,21,22]. On the other hand, bigger brain infarct was observed in ATF3 KO mice with enhanced expression of pro-apoptotic and inflammatory genes [23]. Furthermore, ATF3 has been shown to inhibit NF-κB-triggered inflammation in ischemic organs [3,7,24,25,26]. In addition, bioinformatics analysis suggested that ATF3 may have protective effect on ischemic stroke [26,27]. However, the cellular and molecular mechanism underlying ATF3-acssociated signaling cascade in ischemic brain is largely unknown.

Our previous study with primary cortical neurons revealed the ability for ATF3 to downregulate carboxyl-terminal modulator protein (CTMP), by competing binding with NF-κB, and subsequently reducing oxygen–glucose deprivation (OGD)-induced apoptotic neuronal death [28]. CTMP, an endogenous inhibitor for anti-apoptotic Akt/PKB, through binding to C-terminal of Akt concealed the phosphorylation sites and prevented its activation [29]. However, despite the study in vitro, whether this neuroprotective ATF3→CTMP signaling cascade also exists in the ischemic brain remains to be determined. In the present study with ischemic brain, we showed that ATF3 overexpression could reduce the CTMP level; whereas ATF3 knockout enhanced the CTMP level. Furthermore, in vivo CTMP siRNA knockdown was able to mimic the beneficial effects of ATF3 overexpression, and post-ischemic treatment of either ATF3 overexpression or CTMP knockdown led to decrease in infarct and better functional outcome.

To date, most of the previous studies have been focused on the role of ATF3 against ischemia-induced inflammatory response. For example, in ATF3 KO mouse, increased ischemic brain injury was associated with an increase in inflammatory cell recruitment and the subsequent inflammatory responses [23]. However, whether/how endogenous ATF3 constrains inflammatory cell recruitment and inflammatory gene expression remain to be studied. Recently, using gene ontology (GO) enrichment and KEGG pathway approaches, it was suggested that ATF3 via targeting CCL2-mediated TLR4/NF-κB signaling repressed microglia activation upon cerebral ischemia [26]. However, how ATF3 repressed CCL2 expression was not known. In the present in vivo study and together with our previous in vitro study [28], we uniquely demonstrated that neuronal ATF3 via competing binding with NF-κB downregulates CTMP expression and subsequent neuronal apoptosis upon ischemic brain injury. The above findings implied that ATF3-mediated CTMP downregulation may serve as an endogenous checkpoint retarding the outbreak of ischemia-triggered neuronal apoptosis. In addition, our studies showed that post-ischemic treatment of either Ad-ATF3 or CTMP siRNA could attenuate ischemic brain infarct and improved functional outcome; these therapeutical potentials have not been reported before.

## 2. Results

### 2.1. Induction of ATF3 mRNAs in the Ischemic Cortex after Ischemia-Reperfusion

In the first experiment, a time-course study was carried out to examine the expression of ATF1–7 mRNAs following transient ischemia and reperfusion (Figure 1). RT-PCR was carried out with RNA extracted from the non-ischemic (sham) and ischemic cerebral cortices of rats subjected to MCA occlusion and various reperfusion periods. The primer pairs used for PCR amplification of each ATF isoforms were tested and each resulted in a single cDNA fragment when visualized with ethidium bromide on agarose gel and subsequently sequencing for its identity in GenBank. All seven isoforms of ATF are expressed in the sham cortex, however, relatively low basal levels of ATF3 mRNA were detected (Figure 1A). Apparent induction of ATF3 mRNA was noted at 0.5 h after the onset of MCA occlusion, and expression continued to escalate upon reperfusion, and peaked (~5-fold increase) at 12 h, then gradually subsided with *p* < 0.0001 (Figure 1B,C). A much lower induction (~1.5-fold increase) was noted for ATF4 mRNA. This increase did not start until 0.5 h after the onset of reperfusion, no further increase was observed up to 24 h of reperfusion and then back to basal level with *p* < 0.0001 (Figure 1C). The rest isoforms were not significantly changed. The above semi-quantitative RT-PCR results were further supported by real time quantitative PCR (qPCR) (Figure 1D). As high as ~20-fold increase in ATF3 level was noted (*p* < 0.0001), while the rest ATFs were less than 1.5-fold changes. In agreement with the semi-quantitative RT-PCR analysis, results of qPCR also suggested an important role for ATF3 in ischemic brain injury as compared to other isoforms.

### 2.2. ATF3 Attenuates Brain Infarct upon Transient Ischemic Insult

In order to better understand the neuroprotective role of ATF3 on ischemia-reperfusion challenge, two experimental approaches, namely ATF3 overexpression and knockout (KO), were used. For the overexpression approach, we first evaluated the efficiency of gene expression of adenoviral administration via the intracerebroventricular (icv) route. Here, we infused 1 *×* 10^6^ pfu/10 µL of Ad-ATF3 or Ad-hPGK (untransduced control) into the right lateral ventricles, and determined ATF3 protein levels in the ipsilateral cortices at 12 h to 2 days (2 d) after infection. As compared to Ad-hPGK (2 d), rats injected with Ad-ATF3 augmented ATF3 protein levels in a time-dependent manner, and reached maximal augmentation (~5-fold) at 2 d after administration (Figure 2A). In another set of animals, following 2 d virus infection, rats were subjected to 35 min of MCA occlusion, and ATF3 protein expression determined at 24 h of reperfusion in the ischemic cortices. Under this experimental condition, expression of ATF3 protein in rats infected with Ad-ATF3 (MCAO’) was only ~2-fold higher than the Ad-hPGK control (MCAO’) (Figure 2A). This decrease is likely due to an endogenous induction of ATF3 after ischemic challenge in the Ad-hPGK control. In any case, the results of the above experiments showed that icv infusion of Ad-ATF3 was efficacious in overexpression of ATF3. We then evaluated the effects of pre-ischemic icv Ad-ATF3 administration on infarct volume. Infarct volume in rats receiving Ad-ATF3 at 48 h before MCA occlusion was significantly reduced as compared to the Ad-hPGK control (Figure 2B). Furthermore, when Ad-ATF3 was administered immediately after 35 min of MCA occlusion (namely post-ischemia, at the onset of reperfusion), the infarct volume determined at 24 h of reperfusion was also significantly reduced in the Ad-ATF3 treated group as compared to the Ad-hPGK control (Figure 2D).

For the ATF3 underexpression approach, ATF3 KO mice (a generous gift from Dr. Tsonwin Hai at Ohio State University) were used to test the role of ATF3 on cerebral ischemia. Analysis with RT-PCR and Western blot did not detect ATF3 mRNA or protein in KO brain tissue, despite the fact that a substantial amount of ATF3 mRNA and protein were detected in the wild type (WT) control (Figure 3A). ATF3 KO and WT mice were then subjected to 25 min of MCA occlusion, and infarct volume was measured at 24 h of reperfusion. Under this condition, infarct volume was significantly larger in the ATF3 KO mice as compared to the WT control (Figure 3B).

In order to verify that the exacerbating effect was due to ATF3 deletion, ATF3 was re-introduced back to the ATF3 KO mice by icv infusion of 2 *×* 10^5^ pfu Ad-ATF3 at 48 h prior to 25 min of MCA occlusion, and infarct volume measured at 24 h of reperfusion. Consistent with the above results, infarct volume in KO mice receiving Ad-hPGK was significantly higher than the WT mice receiving Ad-hPGK control (Figure 4A). Furthermore, in agreement with the results of rat studies, infarct volume in WT mice receiving Ad-ATF3 was significantly reduced as compared to the Ad-hPGK control (Figure 4A). In addition, infarct volume in KO mice receiving Ad-ATF3 was also significantly reduced as compared to the Ad-hPGK control (Figure 4A). Furthermore, there was no significant difference in infarct volume between WT mice and KO mice receiving Ad-ATF3 (Figure 4A).

Subsequently, with results showing the ability for ATF3 to reduce infarct volume, experiments were carried out to examine whether this paradigm also resulted in better functional outcome after ischemic insult. Here, mice behavioral ability was assessed by prehensile test (Figure 4B) and beam balance test (Figure 4C,D). All experiments were conducted and recorded at 24 h after MCA occlusion. In the prehensile test, in line with the results of infarct volume, longer latency time (indicating better function) was observed in WT mice than KO mice receiving Ad-hPGK (Figure 4B). In both WT and KO mice, those receiving Ad-ATF3 showed longer latency time than those receiving Ad-hPGK. Furthermore, KO mice receiving Ad-ATF3 had longer latency time than WT mice receiving Ad-hPGK (Figure 4B). There was no difference in latency time between WT mice and KO mice receiving Ad-ATF3 (Figure 4B). Correspondingly, longer holding time (indicating better function) were observed in WT mice than KO mice that receiving Ad-hPGK in the beam balance test; however, no significant changes were observed in the neurological score (Figure 4C,D). In addition, mice receiving Ad-ATF3 had a lower neurological score and longer holding time than those receiving Ad-hPGK in KO mice; however, only a longer holding time was noted in WT mice (Figure 4C,D). Although there was no significant difference in wild-type mice subjected to Ad-hPGK vs. Ad-ATF3 (*p* = 0.0827), this is very close to significance (Figure 4C). In addition, we observed a significant difference between WT-Ad-ATF3 and KO-Ad-hPGK strengthening the beneficial effect of exogenous ATF3 (Figure 4C). There was no difference in neurological score or holding time between WT mice and KO mice receiving Ad-ATF3 (Figure 4C,D).

### 2.3. ATF3 Led to Downregulation of CTMP and Upregulation of p-Akt(473) Expression In Vivo

Our previous in vitro study with hypoxic insult on primary cortical neurons indicated that ATF3 through binding to the CTMP promoter hindered NF-κB binding and inhibited CTMP transcription, and subsequently Akt inactivation and neuronal apoptosis [28]. In order to further study the molecular mechanism underlying reduction of infarct volume by AFT3, Western blot analysis was used to examine whether the neuroprotective action of the ATF3→CTMP signal pathway also exists in the ischemic brain in vivo. In this study, rats were subjected to icv Ad-ATF3 or Ad-hPGK infusion at 48 h prior to 35 min MCA occlusion and 24 h reperfusion, and levels of apoptosis related proteins were analyzed in the ischemic cortices. As compared with the Ad-hPGK control, rats infected with Ad-ATF3 showed higher levels of ATF3, anti-apoptotic proteins (i.e., Bcl-2 and p-Bad), and p-Akt(473), but lower levels of pro-apoptotic proteins (i.e., cleaved caspase 9 and 3) and CTMP (Figure 5A,C). On the contrary, when ATF3 KO and WT mice were subjected to 25 min MCA occlusion and 24 h reperfusion, analysis of proteins showed higher levels of pro-apoptotic proteins (i.e., cleaved caspase 9 and 3 and PARP1) and CTMP, but lower levels of anti-apoptotic proteins (i.e., Bcl-2 and p-Bad) and p-Akt(473) in the KO mice as compared with WT control (Figure 5B,D). Taken together, the aforementioned in vivo data are in line with the previous data from primary neuronal culture in vitro [28], suggesting the neuroprotective action of the ATF3→CTMP signal pathway in ischemic brain in vivo.

### 2.4. Post-Ischemic CTMP siRNA Treatment Ameliorates Brain Infarct and Neurological Deficits

To determine the impact of CTMP downregulation on ischemic brain injury, CTMP mRNA expression profile in the ischemic cortex was examined in rats subjected to MCA occlusion and various reperfusion periods as illustrated in Figure 1A. CTMP mRNA was rapidly induced upon MCA occlusion, reaching a plateau (~2.5-fold) at 30 min of reperfusion and the plateau remained up to 1.5 h, and then gradually subsided before returning to basal level at 336 h (14 days) (Figure 6A). We believe that this early surge in CTMP expression may serve as a master switch to initiate post-ischemic damage. It is interesting to note that expression of ATF3 was going up at 4 h reperfusion, while CTMP level was going down.

To further assess the impact of the ATF3→CTMP signaling cascade in the setting of cerebral ischemic injury, CTMP siRNA was icv infused after 35 min MCA occlusion, and infarct volume was determined after 24 h reperfusion. Results indicated dose-dependent reduction in infarct volume with increasing amount of CTMP siRNA as compared with scRNA control (Figure 6B). CTMP protein level was significantly reduced in the ischemic cortices of rats transfected with CTMP siRNA (75 pmol) as compared with those transfected with scRNA (Figure 6C). Under this condition, the p-Akt(473), but not p-Akt(308), level was increased upon CTMP knockdown, and this was accompanied by increase in p-Bad and decrease in cleaved caspase 3 (Figure 6C). Taken together, these findings support the efficacy of CTMP siRNA in downregulating CTMP and subsequently attenuated the pro-apoptotic drive under cerebral ischemia. Furthermore, the ATF3 level was not altered by CTMP siRNA (Figure 6C), in agreement with the notion that ATF3 is upstream of CTMP. Importantly, post-ischemic CTMP siRNA treatment, as late as 2 h after the onset of reperfusion, could also significantly reduced infarct volume (Figure 6D), suggesting the possibility for CTMP as a therapeutic target for stroke.

Lastly, based on the Bederson’s postural reflex test, post-ischemic CTMP siRNA treatment could also lead to better functional outcome up to 14 days of reperfusion (Figure 7A,B). The protective effect was further supported by the sustained smaller infarct volume as determined by the T2-weighted MR images at 14 days of reperfusion (Figure 7C,D). The advantages of using MR images are (1) to document the temporal evolution of ischemic lesions over time, (2) to reduce the individual various of animals and consequently lower the number of animals used in experiments.

## 3. Discussion

In the present study, biochemical and molecular techniques were used to support the existence of the neuroprotective ATF3→CTMP signal pathway in the ischemic brain in vivo. In addition, our studies demonstrated that post-ischemic treatment of either Ad-ATF3 or CTMP siRNA could attenuate ischemic brain infarct and improve functional outcome. The finding of this neuroprotective ATF3→CTMP signal pathway not only provides new therapeutical targets but also broadens therapeutical options for treating stroke.

Although ATF3 is a recognized neuronal marker of nerve injury, other ATF proteins are also involved in stress response and neurodegeneration [6,30,31]. In order to decipher the complex signal cascades involving ATF3 in the ischemic brain, we first examined the expression profile of ATF isoforms. Our results clearly revealed that ATF3 is the major isoform in response to ischemic challenge. ATF3 mRNA was rapidly induced upon MCA occlusion and carried on increasing after reperfusion. These results are similar to the induction pattern of ATF3 mRNA observed earlier in hypoxic primary neurons [28]. In addition to the marked induction of ATF3 mRNA, a relatively small but significant induction of ATF4 mRNA was also observed after reperfusion, which paralleled the temporal expression of ATF3 mRNA. Despite ATF2, ATF4, and ATF6 having been shown to induce the expression of ATF3, it is very likely that ATF4 may be responsible for the upregulation of ATF3 in the ischemic brain [6,30,31]. In agreement with our observation, results of bioinformatics analysis also imply that ATF3 and ATF4 might affect the development and progression of ischemia-reperfusion injury [32]. Nevertheless, the exact role of ATF4 remains to be investigated.

With the gain-of-function approach, administration of Ad-ATF3 48 h before transient MCA occlusion was shown to attenuate brain infarct. This result is in line with the previous report that administration of rAAV-ATF3 three weeks before permanent MCA occlusion could reduce brain damage [20]. In the present study, we further demonstrated that mice receiving Ad-ATF3 not only showed smaller infarct but also better functional outcome. In addition, administration of Ad-ATF3 immediately after transient MCA occlusion also significantly reduced infarct volume; these result further suggest a therapeutical potential for ATF3. In agreement with the loss-of-function approach described by Wang et al. [23], our study here also showed bigger infarct volume in ATF3 KO mice after transient MCA occlusion. However, traditional genetic knockout approach may be limited in scope due to the presence of regions of genetic variability (“passenger” or “flanking” genes) and/or activation of unexpected compensatory mechanisms [33]. To rule out these complications, we performed an in vivo complementation phenotype rescue study by re-introducing ATF3 back to ATF3 KO mice. Under this condition, complete rescue of infarct volume and neurological deficits were observed in KO mice receiving Ad-ATF3, further supporting the hypothesis that ATF3-deletion is responsible for the bigger infarct volumes and worse functional outcome.

Our previous in vitro study with primary cortical neurons indicated that ATF3 could reduce OGD-induced apoptotic neuronal death by inhibiting CTMP transcription and indirectly upregulating anti-apoptotic p-Akt(473) [28]. To address the molecular mechanism underlying the reduction of brain infarct by AFT3, we showed that ATF3 overexpression led to a lower level of CTMP, and meanwhile elicited higher levels of p-Akt(473) and anti-apoptotic proteins in the ischemic brain. On the contrary, ATF3 KO resulted in an increasing level of CTMP and pro-apoptotic proteins, while a decreasing level of p-Akt(473) in the ischemic brain. Taken together, the aforementioned data suggested the existence of neuroprotective ATF3→CTMP signal pathway in the ischemic brain in vivo.

Through binding to the C-terminal of Akt and thus concealing its phosphorylation sites, CTMP could serve as an endogenous inhibitor for anti-apoptotic Akt/PKB [29]. The pathophysiological significance of the ATF3-CTMP signaling cascade is strengthened by recent studies showing the critical role of CTMP in ischemic neuronal death [34,35,36]. The 4-vessel occlusion global ischemia is known to cause delayed death of hippocampal CA1 pyramidal neurons, and this condition was accompanied by a sustained increase in CTMP level. Depletion of CTMP by lentiviral-mediated CTMP miRNA 14 days prior to 4-vessel occlusion prevented neuronal death in the hippocampal CA1 region [34]. Administration of CTMP siRNA 24 h prior to MCA occlusion also reduced infarct volume and improved neurological deficits [35]. In addition, age-related upregulation of CTMP contributed to bigger ischemic infarct in older rats [36].

In the present study, we showed that both ATF3 and CTMP level were rapidly induced in the ipsilateral cortex following MCA occlusion. However, unlike ATF3 mRNA whose level was marked and prolonged, increasing after reperfusion, CTMP mRNA level was only moderately and transiently increased after reperfusion. While the level of CTMP began to descend, the ATF3 level was still at its ascending phase. This temporal negative association of ATF3 and CTMP expression was also noted in primary neurons after OGD-reoxygenation [28]. Together, the above information implied that ATF3-mediated CTMP downregulation may serve as an endogenous checkpoint retarding the outbreak of ischemia-triggered death signaling.

In addition, the pivotal role of CTMP in ischemic brain injury is further reinforced by the findings that post-ischemic CTMP knockdown with CTMP siRNA treatment could also reduce infarct volume. The neuroprotective action of CTMP siRNA was accompanied by an increase in level of p-Akt(473), but not p-Akt(308), supporting the action of down-regulating CTMP to reduce its inhibition of Akt phosphorylation at serine 473. The fact that CTMP knockdown did not alter ATF3 expression is in support of the contention of the sequential ATF3→CTMP signaling cascade. In addition, besides smaller brain infarct, post-ischemic CTMP siRNA treatment also resulted in better functional outcome. There is growing consensus that siRNA-based therapeutics represent a new way to treat human diseases that are otherwise not drug treatable with existing medicines [37]. Together, these studies support a novel finding and suggest therapeutic potential for both ATF3 overexpression and CTMP knockdown in treating stroke disease.

## 4. Materials and Methods

### 4.1. Stroke Model

The focal cerebral ischemia model entailing 3-vessel occlusion was performed as described [38]. Long–Evans rats, 7- to 8-week-old male of 300 to 350 g body weight, were purchased from The National Laboratory Animal Center (NLAC; Taipei, Taiwan) for our studies. In brief, male Long–Evans rats were anesthetized with chloral hydrate (360 mg/kg body wt, ip), and the right middle cerebral artery (MCA) was ligated reversibly with a 10- 0 suture and both common carotid arteries (CCAs) occluded also reversibly with aneurysm clips. Unless otherwise specified, the 3-vessel occlusion was held for 35 min, and then the suture and the aneurysm clips were released with restoration of blood flow in the 3 arteries verified. While under anesthesia, the rectal temperature was monitored and maintained at 37.0 ± 0.5 °C using a homeothermic blanket (Harvard, Cambridge, MA, USA). Animals were kept in an air-ventilated incubator at 24.0 ± 0.5 °C and sacrificed under anesthesia at variable intervals after focal cerebral ischemia. Brains were removed and ischemic and contralateral cerebral cortices were isolated and frozen. The infarct area was delineated by 2,3,5-triphenyltetrazolium chloride (TTC) and the infarct volume was measured as previously described [38]. The same 3-vessel occlusion model was used in mouse as reported earlier [39]. The ATF3 knockout (KO) mice were kindly provided by Dr. Tsonwin Hai at Ohio State University [5]. ATF3 KO mice with C57BL/6 background were backcrossed for at least 10 generations in our institutional SPF animal facility before experiments [28]. Male KO and littermate WT control mice, 8- to 10-week-old of 20 to 30 g body weight, were subjected to 3-vessel occlusion for 25 min. All procedures were performed in accordance with the Public Health Service Guide Approved Procedures for the Care and Use of Laboratory Animals, and approved by the Academia Sinica Animal Studies Committee (IACUC) (protocol IDs: 13-01-494 & 20-12-1590 to TN Lin). All virus and siRNA treatments were performed by an investigator blinded to the surgical groups. The number of animals used in each experimental group were inserted into figures, each dot represented a data point from an individual animal.

### 4.2. RNA Isolation, Reverse Transcription (RT), Semi-Quantitative Polymerase Chain Reaction (PCR) and Real Time Quantitative PCR (qPCR)

RNA isolation was performed as previously described [40]. RNA concentrations were determined by NanoDrop 2000 (Thermo Fisher Scientific, Waltham, MA, USA), with ratio of 260/280 ≧ 2.0. RNA quality and DNA free was verified with RNA gel analysis. Total RNA (4 ug) was incubated with RevertAid™ H Minus First Strand cDNA Synthesis Kit (Cat #K1631; Fermentas, Vilnius, Lithuania). The reaction mixture was incubated at 65 °C for polydT oligomer annealing and then extension in buffer, dNTP, reverse transcriptase, and RNase inhibitor in a final volume of 20 μL at 42 °C for 1 h and then 70 °C for 5 min to inactivate the enzyme. Finally, a total of 80 uL DEPC treated water was added to the reaction mixture before storage at −80 °C. One or 2 μL of the RT reaction solution was used in the semi-quantitative PCR reaction. PCR was carried out in a 25 μL final volume containing 0.2 mM dNTP, 0.1 μM of each primer and 1 unit of Tag polymerase (Cat # M0273S; NEB, Ipswich, MA, USA). The mixture was subjected to PCR amplification for 25–30 cycles and incubated at 72 °C for 10 min then cooled to 4 °C (PE, Norwalk, CT, USA). Primers for rat-ATF3(NM_012912) F: 5′-GAGGGCCTGCGGTGACTAC-3′, R: 5′-CTGGCCATTGGACAACTTCA-3′ (301 bp); mouse-ATF3(NM_007498) F: 5′-CTGCCATCGGATGTCCTCTGC-3′, R: 5′-CCGTGCCACCTCTGCTTAGCTC-3′ (400 bp); rat-CTMP(NM_001025017) F: 5′-GTGCCATTGCAACCATCATC-3′, R: 5′-GTCAGAGGCTGCGCTGAAG-3′ (400 bp); mouse-CTMP F: 5′-TCTCTGGCTGTATTAGGGAACACA-3′, R: 5′-GGTAGAAGCCAGGAGAGAGTTCCT-3′ (305 bp); rat-ATF1(NM_001100895) F: 5′-TGCCTGGAGAACCGAGTTG-3′, R: 5′-TCCTTGAGCATCTCTCCTTTGG-3′ (300 bp); rat-ATF2(NM_031018) F: 5′-ACCAGGCCCATTTCCTCTTC-3′, R: 5′-TCTTCGACGGCCACTTGTATT-3′ (400 bp); rat-ATF4(NM_024403) F: 5′-TGTTGGAGAAAATGGACCTGAAA-3′, R: 5′-CACTTCCCAGCTCTAAACTAAAGGA-3′ (300 bp); rat-ATF5(NM_172336) F: 5′-ACCGCAAGCAAAAGAAGAGAGA-3′, R: 5′-ATGCACAGGGAAGTGAAATGG-3′ (300 bp); rat-ATF6(NM_001107196) F: 5′-CTTCGAGGCTGGGTTCATAGA-3′, R: 5′-ATGGGTGGTAGCTGGTAATAGCA-3′ (300 bp); rat-ATF7(NM_001108115) F: 5′-CATCCAACAGGCAAATAGGATCT-3′, R: 5′-TGCCTACCACCATTCCACAA-3′ (300 bp); β-actin(NM_031144) F: 5′-CATCCGTAAAGACCTCTATGCCAAC-3′, R: 5′-CAAAGAAAGGGTGTAAAACGCAGC-3′ (304 bp).

Quantitative real time PCR (qPCR) was performed on QuantStudio™ 5 Real-Time PCR System (Thermo Fisher Scientific, Waltham, MA, USA) per manufacturer’s instruction. Three sets of animals with two replicates, and triplicated for each data point were designed for qRT-PCR analysis. In brief, 0.625 uL of cDNA/RT reaction solution and 0.05 μM primers were added to 10 μL Power SYBR Green Master Mix (Cat #4367659; Thermo Fisher Scientific). Specific primers were designed using Primer Express 3.0.1 software, and listed as following: rat-ATF1 F: 5′-CAGATGACCCCCAACTCAAAA-3′, R: 5′-CCCGGGCAGCTTCTCTGT-3′ (60 bp); rat-ATF2 F: 5′-AGCTGCTTTGACCCAGCAA-3′, R: 5′-CCACTGCCGTGGCCTTT-3′ (53 bp); rat-ATF3 F: 5′-GCAGGCAGGAGCATCCTTT-3′, R: 5′-CCGGGATGGTAAGGCCTAA-3′ (56 bp); rat-ATF4 F: 5′-TCGATGCTCTGTTTCGAATGG-3′, R: 5′-CAACGTGGCCAAAAGCTCAT-3′ (61 bp); rat-ATF5 F: 5′-AGGTGACGGCTTCTCTGATTG-3′, R: 5′-GAGGAAGGAGGGCTGTGAAGT-3′ (59 bp); rat-ATF6 F: 5′-TTCTCTGATGGCCGTGCAT-3′, R: 5′-TGAAGATGACCCACAGAACCAA-3′ (64 bp); rat-ATF7 F: 5′-GGACCACCTGGCAGTTCATAA-3′, R: 5′-GGCCGGGCCGAACTT-3′ (58 bp); β-actin F: 5′- AAGGCCAACCGTGAAAAGAT-3′; R: 5′-GTGGTACGACCAGAGGCATAC-3′ (110 bp). Data were normalized using β-actin as an internal control and relative mRNA expression change between 2 groups was calculated by 2^−ΔΔCt^ method.

### 4.3. Western Blot and Immunoprecipitation Analysis

Western blot analysis was performed as described [41]. In brief, samples were homogenized or lysed with ProteoJET™ Mammalian Cell Lysis Reagent (Cat #K0301) and protease inhibitor cocktail (Roche, Basel, Switzerland). An equal amount of proteins (20 µg) was applied to 10% SDS-polyacrylamide gels and electrophoresed. Separated proteins were electroblotted onto Hybond-P:PVDF membranes (Amersham, Piscataway, NJ, USA). The membranes were blocked in TBST buffer containing 20 mmol/L Tris-HCl, 5% non-fat milk, 150 mmol/L NaCl, and 0.05% Tween-20, at pH 7.5, for 1 h at room temperature. Blots were incubated with antibodies against ATF3 (sc-188, 1:500), β-Actin (sc-81178, 1:8000) [Santa Cruz Biotechnology, Inc., Dallas, TX, USA]; cleaved caspase 3(#9661) & 9(#9509), Bcl-2(#2876), cleaved PARP1(#9544), Akt (or PKB)(#9272), p-Akt(Ser473)(#4058), p-Akt(Thr308)(#9275), CTMP(#4612), p-Bad(Ser136)(#4366), Bad(#9292) (Cell signaling Technology, Inc., Danvers, MA, USA; 1:1000). Protein bands were visualized by an enhanced chemiluminescence system (Cat #32106, Pierce, Rockford, IL, USA). Intensity of protein bands was quantified by ImageJ software (Rawak software company, Berlin, Germany). Three rats (*n* = 3) for each group and each band represents a data point that was extracted from an independent brain sample and no replicates. Each band/image was quantified and normalized by β-actin, as internal control. The relative protein expression change was compared to control.

### 4.4. Preparation of Replication-Defective Recombinant Adenoviral Vectors

Viral vectors were prepared as previously described [42,43]. We constructed in the replication-defective recombinant adenoviral (rAd) vector a human phosphoglycerate kinase (hPGK) promoter to drive ATF3 (Ad-ATF3) or a hPGK promoter alone to serve as the control (Ad-hPGK). Replication-defective rAd vectors were generated by homologous recombination and amplified in HEK293 cells. rAd stocks were prepared by CsCl gradient centrifugation and stored at −80 °C. Viral titers were determined by a plaque-assay method. HEK293 cells were infected with serially diluted viral preparations and then overlaid with low melting-point agarose after infection. Numbers of plaques formed (pfu) were counted within 2 weeks.

### 4.5. Intracerebroventricular (icv) Infusion of Ad-ATF3 and CTMP siRNA

The procedure was performed as previously described [39,43]. Briefly, anesthetized rats or mice were placed in a stereotaxic apparatus; 10 or 5 μL of artificial cerebrospinal fluid containing rAd at 1 *×* 10^6^ (rat) or 2 *×* 10^5^ (mouse) pfu was infused into the right lateral ventricle at a rate of 2 or 1 μL/min at the following coordinates: Anterior, 2.5 or 1.5 mm caudal to bregma; Right, 2.8 or 1.5 mm lateral to midline; and Ventral, 3.0 or 1.5 mm ventral to dural surface for rat or mouse, respectively. To delineate the distribution of the transgene expression, we infused Ad-green fluorescence protein (GFP) into the right lateral ventricle for 72 h and GFP was visualized under microscopy. GFP was detected in the lining of ependymal cells and cells surrounding the right ventricle in all 8 coronal brain slices but not in the left ventricle, which had been published in the supplement data Figure S–I of a previous publication [44]. Periodic confirmation of proper placement of the needle was performed with infusion of fast green (Figure 2E and Figure 3D). Similar protocol was used to infuse 10 μL of artificial cerebrospinal fluid containing 25, 50, or 75 pmol CTMP siRNA into the right lateral ventricle at a rate of 2 μL/min. The concentrations of CTMP siRNA selected here were based on our previous experience. Specific CTMP siRNA and scRNA were purchased from MDBio Inc. (Taipei, Taiwan).

### 4.6. Behavioral Assessment

After MCA occlusion, rats exhibited unilateral forelimb weakness and flexion with reduced resistance when suspended by the tail and with applied pressure on their shoulders. Bederson’s test was used to evaluate neurological deficits based on the postural reflexes before MCA occlusion and 2- and 14-days after MCA occlusion by investigators blinded to the study conditions. Rats were scored based on the following criteria: grade 5 (normal); grade 4 (forelimb flexion and no other abnormalities); grade 3 (reduced resistance to lateral push toward the paretic side, and forelimb flexion); grade 2 (same behavior as grade 3, with circling toward the paretic side when pulling the tail on the table); grade 1 (same behavior as grade 2, with spontaneous circling); grade 0, no activity [40,45].

Behavioral ability in mice was assessed by prehensile test and beam balance test, with conductions recorded at before and post MCA occlusion day 1. In the prehensile test, mice were placed onto the round plate (3, 4, or 5 cm in diameter and set horizontally and suspended 60 cm above a foam pad to protect against fall injuries) to evaluate their balance ability and the power of the forepaw. The time until falling was tested two times and averaged. In the beam balance test, a wooden rectangular beam, 80 cm in length and 1.2 or 2.1 cm in width, was set horizontally and suspended 60 cm above a foam pad. Mouse neurological performance was assessed on a 6-point scale (from good to poor performance): (0), balance with steady posture on the beam (stay > 60 s or walk into cage box); (1), grasping side of beam (stay > 60 s); (2), one limb falling down (stay > 60 s); (3), two limbs falling down (stay > 40 s); (4), try to balance but fall off (stay > 20 s); (5), stay for > 10 s; and (6), cannot balance on the beam (stay < 10 s). The behavior was recorded by video and tested two times, and the results were averaged. All the experiments were conducted by a trained investigator blinded to the experimental conditions [41,46,47].

### 4.7. Magnetic Resonance Imaging (MRI) Experiment

MRI studies were performed on a 7-T PharmaScan 70/16 MR scanner (Bruker Biospin GmbH, Ettlingen, Germany) with an active shielding gradient (30 G/cm in 80 μs). Rats were initially anesthetized with 5% isoflurane and maintained with 1.5 to 2.0% isoflurane at 2 L/min air flow. Rats were allowed to breathe spontaneously throughout the experiments. Rats were placed in a prone position and fitted with a custom-designed head holder inside the magnet, as previously described [48]. Images were acquired using a 72-mm birdcage transmitter coil and a separate quadratic surface coil for signal detection. Multi-slice axial FES T2WI were acquired with a field of view of 3.0 *×* 3.0 cm^2^, a matrix size of 256 × 128 with zero-filling to 256 *×* 256, repetition time = 4500 ms, echo time = 70 ms, bandwidth = 50 kHz, slice thickness = 1 mm, number of averages = 8, and the total acquisition time is 9 min 36 s. T2WI was assessed by investigators blinded to the study conditions at 14 days after ischemia.

### 4.8. Statistical Analysis

Data were first checked with Shapiro–Wilk test and Levene’s test for normality and homogeneity, respectively; and the suitability for ANOVA analysis. If not, Kurstal–Wallis test, post-hoc Dunn’s test and linear regression were used to compare the expression of protein, mRNA and Bederson’s test using SAS 9.4 and R 4.1 software. In addition, the Bonferroni test was applied to minimize the multiple testing issue. *p* < 0.05 was considered significant.

## 5. Conclusions

In summary, the present proof of principle studies advocates the in vivo causal link between ATF3 and CTMP as key players in the regulation of ischemic brain injury and functional outcome. Consequently, delineating the ability for ATF3 to trans-repress CTMP will help to broaden insights into the development of more effective therapies to restrain brain damage caused by ischemic stroke.

## Figures and Tables

**Figure 1 ijms-24-02306-f001:**
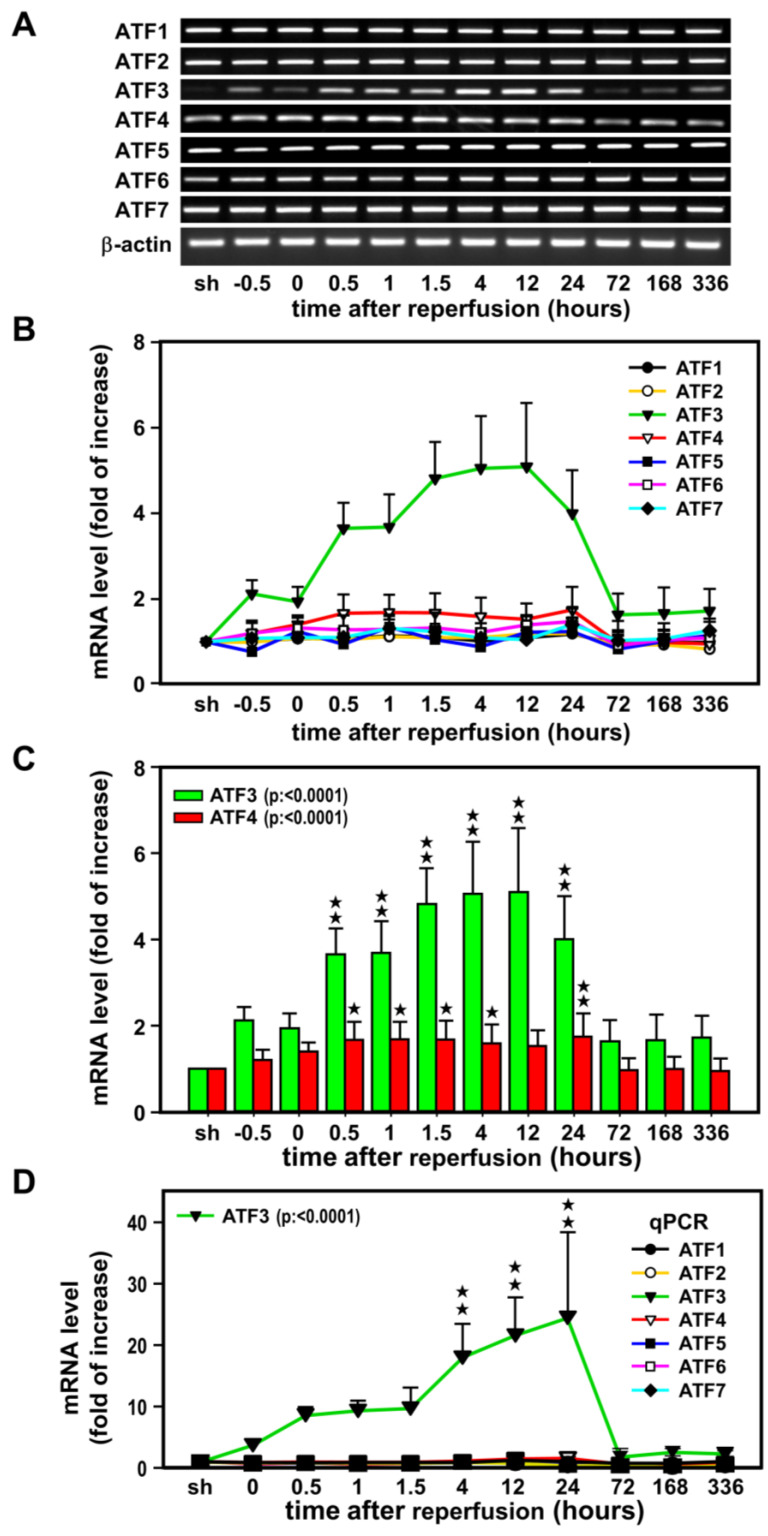
A time-course study of the expression of ATF1–7 mRNAs following transient ischemia. Representative RT-PCR results of ATF1–7 mRNAs expression in the ipsilateral cortex of rats sub-jected to 60 min of MCA occlusion (**A**). Three rats (*n* = 3) for each group and each band represents a data point that was extracted from an independent brain sample and no replicates. The ATF1–7 mRNA bands were quantified and normalized with β-actin mRNA (internal control). The value obtained from the sham-operated controls (sh) was arbitrarily defined as 1 (**B**). ATF3 and ATF4 mRNAs were the only two isoforms showing significant increase (**C**). The real time qPCR results of ATF1–7 mRNAs expression in the ipsilateral cortex are shown in panel (**D**). qPCR were performed in three rats (*n* = 3), 2 replicates and triplicate for each point. Values: –0.5 and 0 represent 30 and 60 min after the onset of ischemia; 0.5, 1, 1.5, 4, 12, and 24 denote the duration of reperfusion in hours after 60 min of ischemia. Data are expressed as mean ± SD (*n* ≧ 6). ★ *p* < 0.05 and ★★ *p* < 0.01 vs. sh. ATF: activating transcription factor.

**Figure 2 ijms-24-02306-f002:**
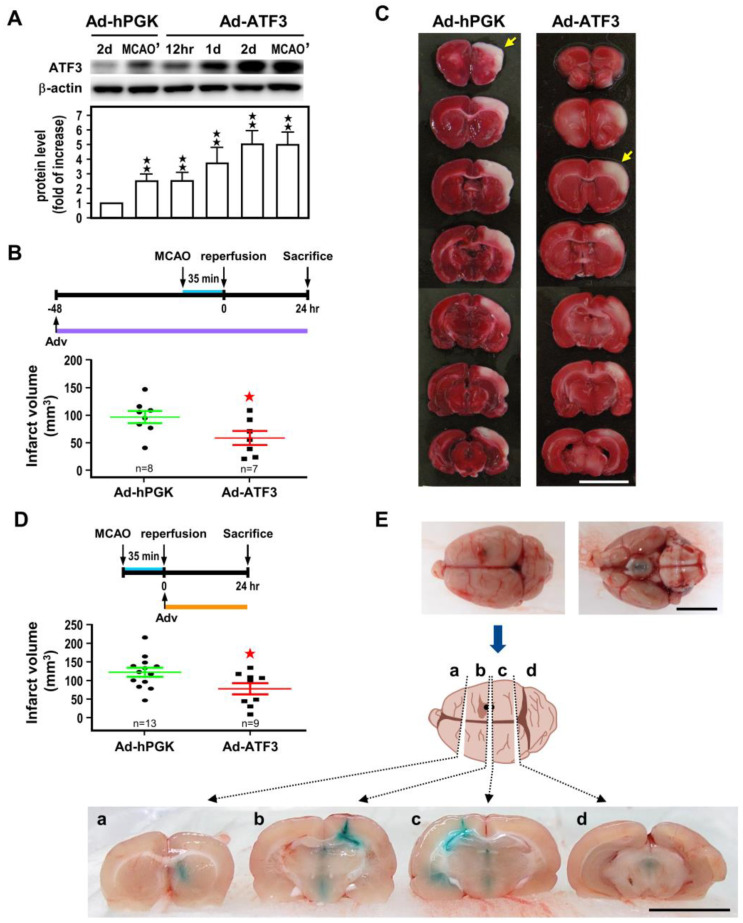
ATF3 overexpression attenuates ischemic brain injury. Rats were subjected to icv infusion of Ad-ATF3 or Ad-hPGK control at 48 h/2 d prior to (**A**–**C**) or right after (**D**) 35 min of ischemia and infract volume had been determined at 24 h reperfusion. Representative Western blots of ATF3 and β-actin proteins in rat brain infected with adenoviral vectors and densitometric analysis (**A**). Three rats (*n* = 3) for each group and each band represents a data point that was extracted from an independent brain sample and no replicates. Symbols 12 h, 1 d, and 2 d represent 12 h, 24 h, and 48 h after adenovirus infection, respectively; and MCAO’ represents after 2 d infection rats were subjected to 35 min of MCA occlusion (MCAO) and 24 h reperfusion (*n* = 3). Brain infarct volume in rats treated with Ad-ATF3 or Ad-hPGK at 48 h before 35 min of ischemia (**B**) and representative images of brain infarct (**C**). Brain infarct volume in rats treated with Ad-ATF3 or Ad-hPGK immediately after 35 min of ischemia (**D**). Data are expressed as mean ± SD (*n* ≧ 6). ★ *p* < 0.05 and ★★ *p* < 0.01 vs. control. Periodic confirmation of proper placement of the needle was performed with infusion of fast green (**E**). White area represents infarct area (yellow arrow). (Bars = 1 cm). ATF3: activating transcription factor 3; Ad-hPGK: rAd-carrying hPGK promoter; hPGK: human phosphoglycerate kinase; rAd: replication-defective recombinant adenoviral; Ad-ATF3: rAd-carrying hPGK promoter-driven ATF3 gene; MCAO: middle cerebral artery occlusion.

**Figure 3 ijms-24-02306-f003:**
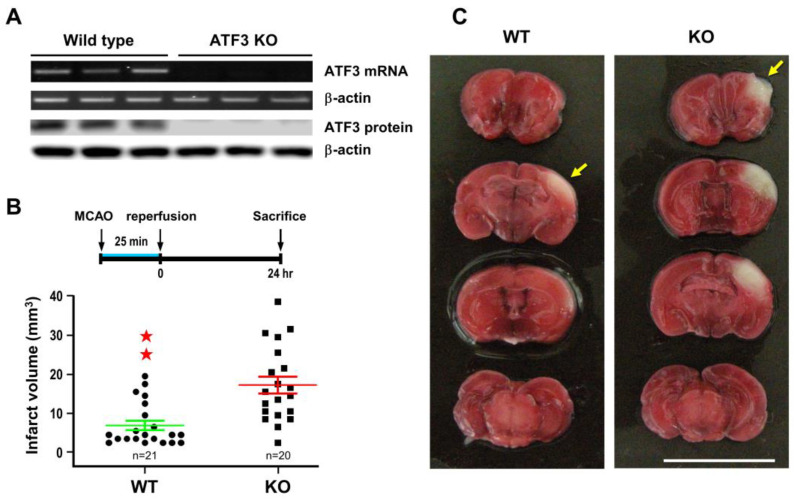
ATF3 knockout (KO) exacerbates ischemic brain injury. ATF3 KO mice were subjected to 25 min of ischemia and infract volume was determined at 24 h of reperfusion. Three wild type and three KO mouse brains were used in RT-PCR analysis; and another set of three wild type and three KO mouse brains were used in Western blot analysis. Each band represents a data point that was extracted from an independent brain sample and no replicates. Representative RT-PCR and Western blots of ATF3 mRNA and protein in the ischemic cortices (**A**). Brain infarct volume in KO and wild type (WT) mice after 25 min of ischemia and 24 h of reperfusion (**B**) and representative images of brain infarct (**C**). Each dot represents a data point from an individual animal. Data are expressed as mean ± SD (*n* ≧ 6). ★★ *p* < 0.01 vs. control. ATF3: activating transcription factor 3; MCAO: middle cerebral artery occlusion.

**Figure 4 ijms-24-02306-f004:**
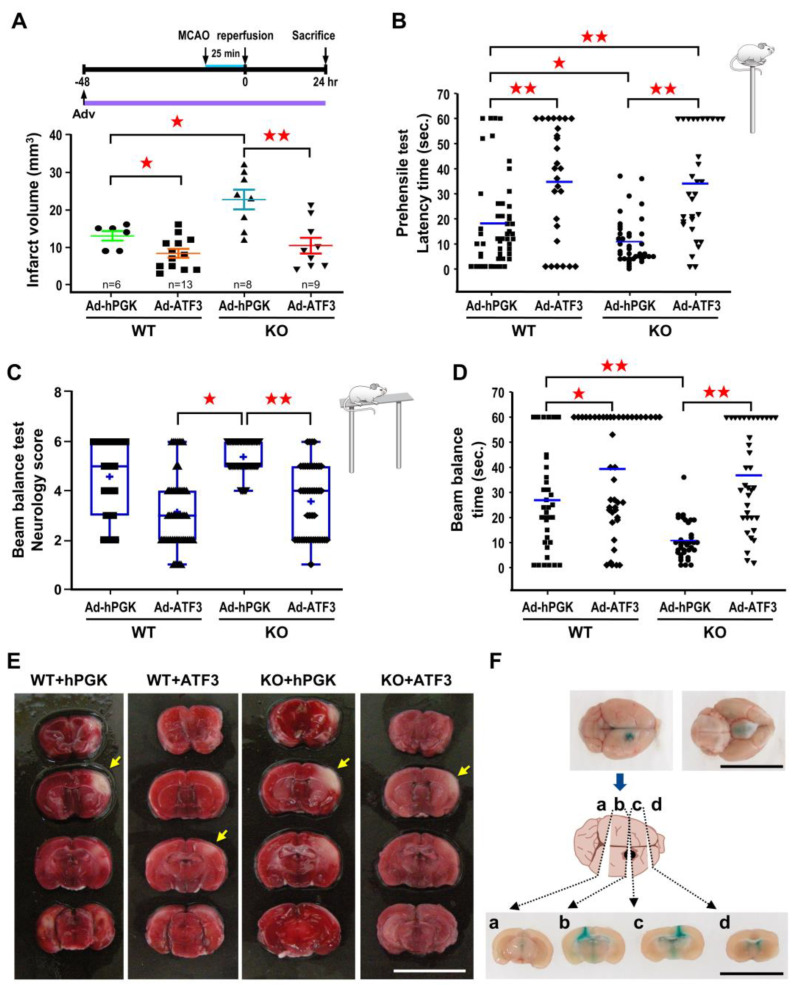
ATF3 overexpression rescues brain infarct and neurological deficits in ATF3 KO mice. ATF3 KO or WT mice were subjected to icv infusion of 2 × 10^5^ pfu of Ad-ATF3 or Ad-hPGK at 48 h prior to 25 min of ischemia and infract volume was determined at 24 h of reperfusion. Brain infarct volumes are quantified in panel (**A**). Each dot represents a data point from an individual animal. Behavior tests based on prehensile test (**B**) or beam balance test (**C**,**D**) were performed at 24 h of reperfusion. Data are expressed as mean ± SD (*n* ≧ 6). ★ *p* < 0.05 and ★★ *p* < 0.01 between groups. Representative images of brain infarct are shown in panel (**E**); white area represents infarct area (yellow arrow). Periodic confirmation of proper placement of the needle was performed with infusion of fast green (**F**). (Bars = 1 cm). ATF3: activating transcription factor 3; Ad-hPGK: rAd-carrying hPGK promoter; hPGK: human phosphoglycerate kinase; rAd: replication-defective recombinant adenoviral; Ad-ATF3: rAd-carrying hPGK promoter-driven ATF3 gene; MCAO: middle cerebral artery occlusion.

**Figure 5 ijms-24-02306-f005:**
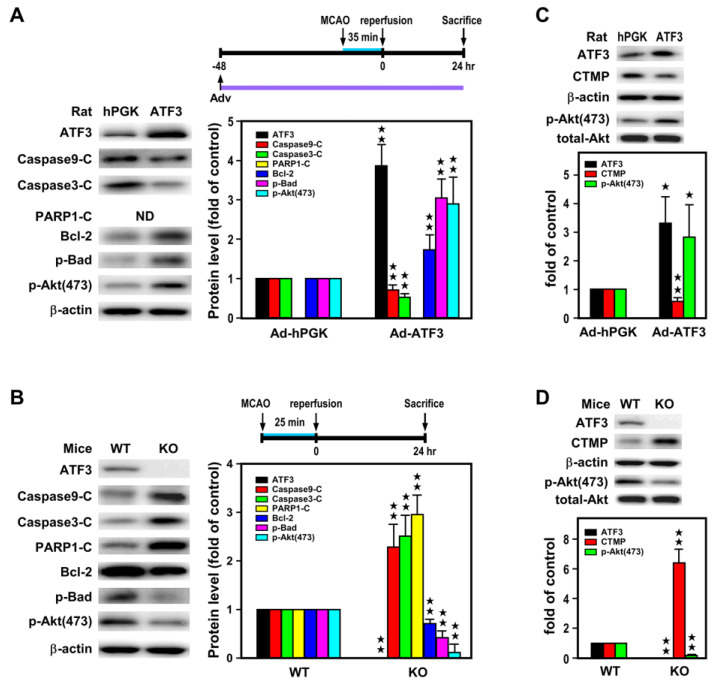
ATF3 overexpression attenuates while knockout aggravates apoptosis in the ischemic brain. Apoptosis-related proteins were measured in ischemic cortices of rats infected with 1 × 10^6^ pfu of Ad-ATF3 or Ad-hPGK and subjected to 35 min of MCAO and 24 h of reperfusion (**A**,**C**). Six rats for each group and each band represents a data point that was extracted from an independent brain sample or in ATF3 KO and WT mice subjected to 25 min of MCAO and 24 h of reperfusion (**B**,**D**). Six WT or KO mice for each group and each band represents a data point that was extracted from an independent brain sample. Apoptosis-related proteins were analyzed by Western blotting. ND indicates not determined. Bar graphs in the adjacent panels show compiled data. Controls were arbitrarily defined as 1 (Ad-hPGK in panel (**A**,**C**); or WT in panel (**B**,**D**)). Data are expressed as mean ± SD (*n* ≧ 6). ★ *p* < 0.05 and ★★ *p* < 0.01 vs. controls. ATF3: activating transcription factor 3; Ad-hPGK: rAd-carrying hPGK promoter; hPGK: human phosphoglycerate kinase; rAd: replication-defective recombinant adenoviral; Ad-ATF3: rAd-carrying hPGK promoter-driven ATF3 gene; MCAO: middle cerebral artery occlusion; CTMP: carboxyl-terminal modulator protein.

**Figure 6 ijms-24-02306-f006:**
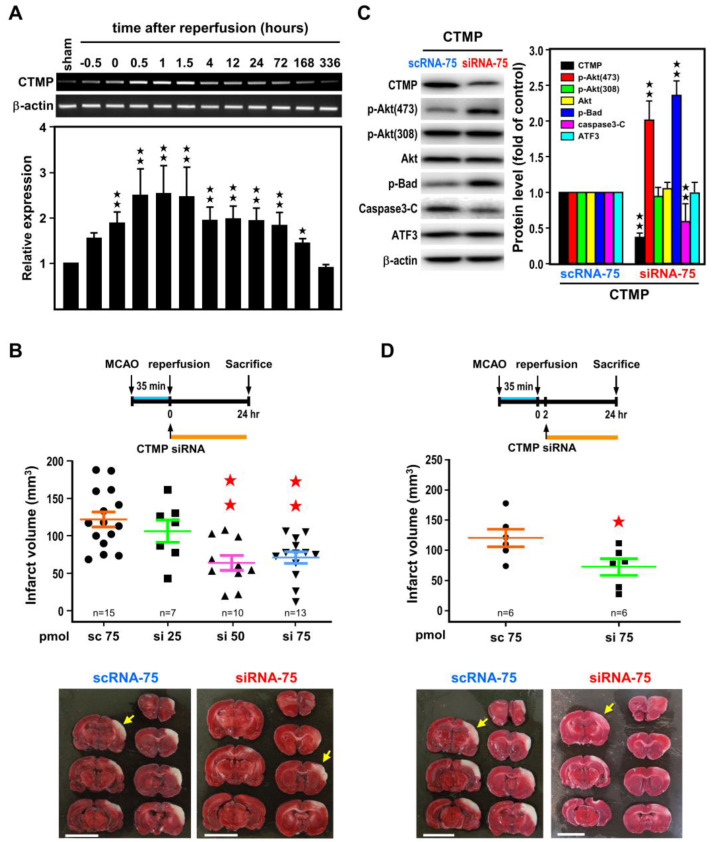
Reduction in ischemic brain injury with post-ischemic CTMP siRNA treatment. (**A**) A time-course study of CTMP mRNA levels in the ischemic cortices of rats subjected to 60 min of MCA occlusion and various reperfusion periods as in Figure 1. Three rats for each group and each band represents a data point that was extracted from an independent brain sample. Representative RT-PCR bands are shown in the upper panel and bar graphs below are compiled data shown as mean ± SD (*n* = 3). (**B**) Rats were subjected to icv infusion of CTMP siRNA at 25, 50, or 75 pmol (si 25, si 50, or si 75, respectively) or 75 pmol of scramble RNA (sc 75) right after 35 min of MCA occlusion, and infarct volume was measured at 24 h of reperfusion. Representative images of brain infarct are shown at the bottom. (**C**) The apoptosis-related proteins were measured at 24 h of reperfusion. Three rats for each group and each band represents a data point that was extracted from an independent brain sample. Representative Western blot bands are shown in the left and bar graphs aside are compiled data shown as mean ± SD (*n* = 3). (**D**) Rats were subjected to icv infusion of CTMP siRNA 2 h after 35 min of MCA occlusion, and infarct volume were measured at 24 h of reperfusion. Representative images of brain infarct are shown at the bottom. Data are expressed as mean ± SD (*n* ≧ 6). ★ *p* < 0.05 and ★★ *p* < 0.01 vs. controls. White area represents infarct area (yellow arrow). (Bars = 1 cm). CTMP: carboxyl-terminal modulator protein; ATF3: activating transcription factor 3; MCAO: middle cerebral artery occlusion; scRNA: scramble RNA; siRNA: small interfering RNA.

**Figure 7 ijms-24-02306-f007:**
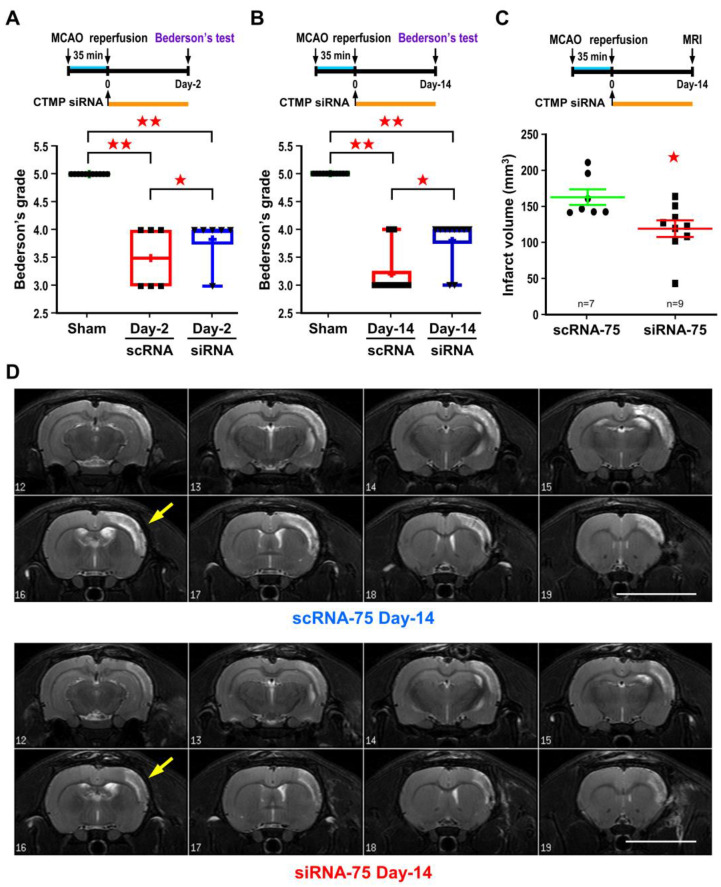
Post-ischemic CTMP siRNA treatment leads to sustained reduction in brain infarct and improves neurological deficit. Rats were subjected to icv infusion of 75 pmol CTMP siRNA right after 35 min of MCA occlusion. Functional outcome based on Bederson’s postural reflex test were measured at 2 and 14 days of reperfusion (**A**,**B**). Rats were then subjected to T2-weighted MRI for infarct volume analysis (**C**), representative T2-weighted MR images of coronal brain sections are shown in panel (**D**). Each dot represents a data point from an individual animal. Data are expressed as mean ± SD (*n* ≧ 6). ★ *p* < 0.05 and ★★ *p* < 0.01 vs. controls. White area represents infarct area (yellow arrow). (Bars = 1 cm). CTMP: carboxyl-terminal modulator protein; MCAO: middle cerebral artery occlusion; scRNA: scramble RNA; siRNA: small interfering RNA.

## Data Availability

Analytic methods, and study materials will be made available on publication of this research article. The data will be available from the corresponding author on reasonable request.

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
