# Peer review of "Activating Transcription Factor 3 Diminishes Ischemic Cerebral Infarct and Behavioral Deficit by Downregulating Carboxyl-Terminal Modulator Protein"

_ijms, 2023, doi:10.3390/ijms24032306_

Round 1

Reviewer 1 Report

The manuscript entitled “Activating Transcription Factor 3 Diminishes Ischemic Cerebral Infarct and Behavioral Deficit by Downregulating Carboxyl-Terminal Modulator Protein” addresses the in vivo role of activating transcription factor 3 (ATF3) in ischemic insult in mice and associated molecular mechanisms. Initially, the authors proved that pre- and post-ischemic administration of ATF3 attenuated brain infarction and neurological deficits in mice. Then, the authors proceeded to some implicated mechanisms. To this end, the authors demonstrated that ATF3 overexpression triggered diminished levels of CTMP and higher levels of p-Akt(473) in the ischemic brain. The authors concluded that ATF3 overexpression and CTMP knockdown may be beneficial targets for stroke management. The current findings are interesting.

 Comments:     

1) What is the novelty of the present work? The authors are advised to elaborate in the introduction section on the sharp differences that highlight the novelty of the present work and how the study is different from previous literature that has already addressed the role of activating transcription factor 3 against cerebral ischemia-triggered neuronal damage in vivo in rodents. What is already described by the authors regarding this issue needs to be further elaborated.

- Hence, the authors are also suggested also to elaborate on how the current study is unique relative to the following articles:

A) Ma et al., 2022 (Protective role of activating transcription factor 3 against neuronal damage in rats with cerebral ischemia, Brain Behav, 2022 Apr;12(4):e2522.  doi: 10.1002/brb3.2522).

B) Wang et al., 20212 (Increased inflammation and brain injury after transient focal cerebral ischemia in activating transcription factor 3 knockout mice, Neuroscience, 2012 Sep 18;220:100-8.  doi: 10.1016/j.neuroscience.2012.06.010).

2) In a separate section in the Material and Methods section, the authors are advised to provide the species, strain, sex, weight, and source of the animals. Please, also provide the number of animals used in each experimental group.

3) How did the authors decide on calculating the sample size per experimental group?

 4) The authors are advised to provide in the ethics statement the ethical approval number.

5) In line 494, how were the concentrations of CTMP siRNA (25, 50, or 75 pmol) selected? Please, can you provide evidence (from literature, for example) that you have not used toxic concentrations in mice?

 6) The qRT-PCR is missing biological (how many samples were used per experimental group) and technical repeat information (whether each sample was repeated during the assay). Moreover, did the authors check the RNA quality with A260/280, and perform an RT negative control to ensure no DNA contamination in the RNA extraction? Please, add these data in the relevant section in material and methods.

7) The author should mention the amount of cDNA (µg) used for qRT-PCR. Please, add these data in the relevant section in material and methods.

8) In the PCR: The authors are advised to add the gene accession number and amplicon size for all target genes. Please, add these data in the relevant section in material and methods.

9) The authors are advised to add the catalog number for the used chemicals, kits, and antibodies.

10) In Western blotting, please described the image analysis (including the number of subjects, the number of images, and the software used).

11) In the statistical analysis section, did the authors check data normality and homogeneity before proceeding to one-way ANOVA? Authors are advised to address this point and add the answers to the comment in the material and methods section.

12) In figures 4C (Beam balance) and figure 7 A, B (Bederson’s test), given the fact that the values for these tests are discrete variables (non-parametric data), ANOVA analysis is not an appropriate test. The authors are advised to analyze the data using Kruskal-Wallis analysis of variance. When statistical significance is obtained, Dunn's test is applied. The authors are advised to redo the statistical analysis for non-parametric data as described. Ideally, the non-parametric data should be presented using boxplot figures.

13) To make all figure legends stand-alone, authors are advised to add the full name of the used abbreviations at the end of each legend. Moreover, figure captions must include information on sample size (n), and an explanation of all abbreviations used in each figure. Authors are advised to address this point and add the answers to the relevant figure legends.

 14) In the legend of figures 3, 5, and 6, the authors are advised to describe the number of replicates used in Western blotting. Moreover, were the data extracted from independent samples? Authors are advised to address this point and add the answers to the relevant figure legends.

Reviewer 2 Report

This is an interesting and generally very well done study on effects of activating transcription Factor 3 and carboxyl-terminal modulator protein on ischemic brain injury. The data are convincing and the conclusions are well supported by the results. I have, however, several minor suggested revisions as follows:

1. In my copy of the manuscript, the legend for Figure 2 is there but there is no Figure 2 with the legend or in the paper anywhere.

2. In Fig 4, it is not necessary to have this huge number of zeros for the p value.

3. The MRI methods are included in the paper as are images for Figure 7 C&D. Although cited in Fig 7 legend, there is no description of the MRI findings in the Results section.

4. Was body temperature monitored during and after stroke? This is an important control.

5. The IACUC protocol approval number needs to be included in the methods.
